# Total Quality and Innovation Management in Healthcare (TQIM-H) for an Effective Innovation Development: A Conceptual Framework and Exploratory Study

Suebsakul Tonjang and Natcha Thawesaengskulthai *

Department of Industrial Engineering, Faculty of Engineering, Chulalongkorn University, Bangkok 10330, Thailand; suebsakul.tj@gmail.com
* Correspondence: natcha.t@chula.ac.th; Tel.: +668-509-64888

**Abstract:** To thrive, an organization must adapt to the fast and constant change in the economic environment caused by an aging society, technological changes, and the pandemic crisis. Innovation becomes important for the adaptation of industries. Healthcare is one of them. Innovation development in hospitals is effective and acceptable when its management is effective and aligns with the healthcare quality context since quality is a philosophy of work in life-related settings. To the best of our knowledge, quality management and innovation management in healthcare have never been integrated. Therefore, this research aimed to create an integrated framework of quality and innovation management in healthcare (TQIM-H). To establish the effectiveness of applying TQIM-H for the development of effective healthcare innovation, this study developed a TQIM-H conceptual framework using multiple methodologies including a literature review, multiple case studies analysis, Delphi study with healthcare experts, Technology Acceptance Model (TAM), and triangulation with an external dataset. We constructed a TQIM-H conceptual framework, consisting of seven dimensions, that can be used in developing innovation projects in hospitals and which agrees with safety and quality principles in hospitals.

**Keywords:** quality and innovation management; healthcare management; healthcare innovation; effective innovation development

## 1. Introduction

The COVID-19 pandemic crisis, aging society, novel health trends, and technological changes affect how humans conduct business globally [1–4]. These factors force organizations to adapt to changes and innovate themselves. One of the adaptations is the adoption of innovation which has suddenly become important in several organizations including healthcare [5–8]. Innovation in product, process, and business models becomes a key factor that can lead to adaptation that serves global needs. Innovation leads to disruption in all activities in all of the hospital sections including telehealth, biosensors and trackers, artificial intelligence, virtual reality, etc. Thus, innovation management is a framework that is critical for healthcare organization management so that the organizations can respond to changes effectively [9–14]. However, several studies demonstrated that healthcare innovation management is difficult since the principle of management does not align with the organizational quality context [15–21]. In healthcare, quality is the foundation and core philosophy, so quality is essential in every step of the process in healthcare organizations to mitigate hazards in patient live [22–25]. Therefore, efficient initiation of innovation in hospitals requires management that agrees with the healthcare quality framework.

The relationship between quality management and innovation management has been researched in multiple industries and the findings have been inconclusive. Quality management can have a negative relationship with innovation management since quality is a strict framework that impedes innovation while innovation can lead to negligence of safety

and quality [26,27]. On the other hand, some researchers proposed that quality supports the innovation development process. High-quality innovation impresses customers and increases working quality and efficiency [28,29].

For the healthcare industry, Tonjang and Thawesaengskulthai (2020) [30] found that quality management has a positive relationship with innovation management in the healthcare industry since innovation maximizes the capability of quality management and clarifies the goal of quality management. At the same time, quality guides the direction of innovation development to be safe, ethical, legal and acceptable. While the two management frameworks have a common goal to serve customer needs, Tonjang and Thawesaengskulthai (2020) only explained the relationship between quality management in healthcare and innovation management in healthcare. The study did not develop the integrated conceptual framework of quality management healthcare and innovation management in healthcare and did not provide the details of the integrated conceptual framework. Therefore, this study aimed to integrate quality management and innovation management so the integrated framework can be used to develop healthcare innovation projects and allow the development to be efficient and agree with the quality background.

To develop the integration framework of healthcare quality and innovation management, we used four methodologies. First, a literature review was used to integrate two management philosophies' factors, resulting in the integrated factors. Second, several case studies were examined to confirm the integrated factors. Third, a Delphi study was carried out to develop the conceptual framework. Forth, The Technology Acceptance Model (TAM) was used to assess the acceptability of the developed framework. The conceptual framework was then tested and validated by using the framework to develop an innovation project in a case study hospital from start to finish. The developed integration framework will be used as a structural management framework whose key dimensions and subfactor components can be used as a guideline for the development of healthcare innovation projects while still allowing hospitals to adhere to the quality principle.

## 2. Research Methodology

This research aimed to develop a TQIM-H conceptual framework by integrating quality management and innovation management in healthcare. Figure 1 demonstrates the concept to develop the integrated conceptual framework of TQIM-H, which is the analysis of two management philosophies, the integration of the two management factors, and the development of an integrated conceptual framework.

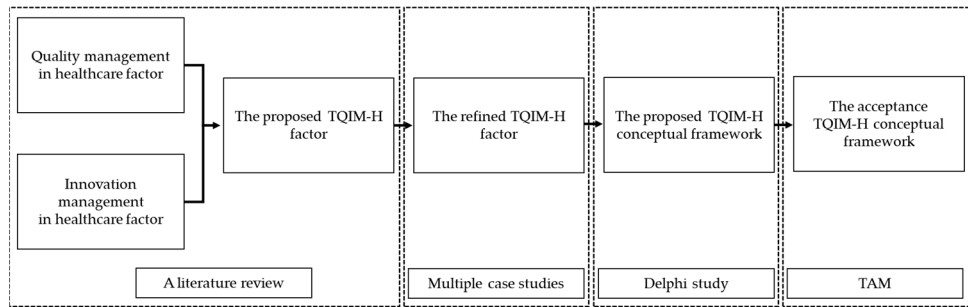

**Figure 1.** The concept to develop TQIM-H.

Four research methods were used to develop the TQIM-H conceptual framework affecting healthcare innovation development (Figures 1 and 2). First, literature reviews were conducted to study the key factors of quality management and innovation management from previous studies. Then, the key factors from the two management principles were integrated. Similar factors were merged and conflicting factors were resolved. This step resulted in the proposed TQIM-H factors. Second, the proposed TQIM-H factors were confirmed through 20 impactful innovation case studies in 18 hospitals to study factors used in these projects that correspond to the factors in TQIM-H. Then, a Delphi study with healthcare experts was used as a tool to develop the TQIM-H conceptual framework. Finally,

the developed conceptual framework was tested for its ease of use and its acceptance level via TAM methodology.

| Methodology | Objective | Result |
|---|---|---|
| 2.1 A literature review | To review and integrate the quality management and innovation management from the previous study | The proposed TQIM-H factors |
| 2.2 Multiple case studies | To refine and confirm the TQIM-H factors | The refined TQIM-H factors |
| 2.3 Delphi with an expert panel | To develop the TQIM-H conceptual framework | The proposed TQIM-H conceptual framework |
| 2.4 Technology Acceptance Model (TAM) | To test the ease of the TQIM-H conceptual framework's use and show the acceptance level of TQIM-H conceptual framework | The acceptance TQIM-H conceptual framework |

**Figure 2.** Steps of process methodology.

## 2.1. A Literature Review

In this process, we studied key factors involved in quality management in healthcare and innovation management in healthcare from the literature. This step was to refine the scope, characteristics, and factors involved in quality management and innovation management in healthcare. Then, we analyzed and integrated the key factors by combining similar factors and resolving conflicting factors. This step yielded the integrated factor of TQIM-H. To study factors involved in TQIM-H, articles related to the integration of two management philosophies emerged in 1985, so we searched articles from 1985 to 2022. SCOPUS, Science Direct, Web of Science, MEDLINE, Google Scholar, and ABI/INFORM databases were used. The full-text articles related to quality management and/or innovation management were reviewed and extracted for information including status, content, and context of research for the integration of quality and innovation management. Then, the quality management factor and innovation management factor from a literature review were analyzed and combined in the following three steps:

Step 1. We reviewed the quality and innovation in healthcare factors. The authors studied and reviewed data collected from a literature review to understand the characteristics and factors of quality management and innovation management in healthcare.

Step 2. We matched the factors of quality and innovation in healthcare. Parallel statements of factors in quality management and innovation management were matched. The statements that agreed were combined and the statements that conflicted were discussed with three healthcare experts to find an agreement. This step yielded the proposed TQIM-H factors.

Step 3. We analyzed the integrated factor. Discrepancies were examined and analyzed before reaching a consensus by a focus group of healthcare experts.

## 2.2. Case Study Analysis

The proposed TQIM-H factors resulting in the previous stage were refined and confirmed through the healthcare innovation projects. The effective 20 case studies which have been established and launched in 18 Southeast Asia hospitals in 2021 were studied and analyzed. The factors used in these projects that correspond to the factors in the proposed TQIM-H were presented to confirm and define the refined TQIM-H framework. The process to study in this area has four steps:

Step 1. We summarized the developed TQIM-H. The proposed TQIM-H factors resulting in the previous stage were summarized as a TQIM-H key factor template.

Step 2. We extended TQIM-H factors to the case studies. The TQIM-H factors template was sent to the effective case studies' project owners.

Step 3. We extracted effective TQIM-H factors via case studies analysis. The case studies' project owners evaluated if their projects had used any TQIM-H key factors. Such information was provided in the TQIM-H factors template.

Step 4. We collected the results from each case study. The case studies' project owners sent the TQIM-H template back to us. In this step, each TQIM-H factor used in each case study was illustrated and counted for its frequency in the usage.

### 2.3. Delphi Study with Expert Panels

In this study, the Delphi technique with healthcare experts was used as a tool to develop the TQIM-H conceptual framework. The benefits of the Delphi technique are the potential for anonymity, the ability to equalize participants, and the ability to remove personality factors from the process [31].

### 2.3.1. Selection of Experts

The selection of experts in the Delphi panel is important for synthesizing new knowledge. Scheele (1975) [32] recommended that the panel must be selected from stakeholders who would be directly affected, experts with relevant backgrounds and experience, and facilitators in the study field. This study used the same 30 healthcare experts for the three rounds of Delphi. Healthcare experts from healthcare organizations were selected based on the required qualification as shown in Table 1. The average working time of the participants in the health sector is 19.8 years.

**Table 1.** Healthcare expert panel criteria.

| Expert Categories | Required Qualification | Number of the Expert Panel |
|---|---|---|
| Academics | More than 5 years of experience in the academic area with a Ph.D. | 6 |
| CEO/Directors | Top management in healthcare and more than 5 years of experience in the healthcare position. | 6 |
| Healthcare quality assurancespecialist | Healthcare quality assurance specialist with healthcare quality certification and more than 5 years of experience in the healthcare position. | 6 |
| Innovation specialist in healthcare | Healthcare innovation specialist and more than 5 years of experience in the healthcare position. | 6 |
| Project development specialist in healthcare | Project manager/technical specialist and more than 5 years of experience in the healthcare position. | 6 |
| Total | | 30 |

A three-round Delphi survey with healthcare experts, to gain further consensus, was used to develop the TQIM-H conceptual framework.

### 2.3.2. Delphi First Round

In the first round, the study started with refining the TQIM-H factors through in-depth interviews with 30 healthcare experts using a Likert scale questionnaire. Then, the questionnaire's results were analyzed by importance and performance analysis (IPA). The process of the first round contains four septs including:

Step 1. We presented TQIM-H factor to expert panels. The TQIM-H key factors from the literature review and case study analysis were presented to the experts so that they were familiarized with the integrated factors that were developed from the literature review and case studies analysis.

Step 2. We interviewed the experts about the TQIM-H via questionnaire. An in-depth interview with healthcare experts provided opinions that were used to develop TQIM-H. The opinions on the importance and performance of each TQIM-H key factor were expressed as scores ranging from 1 to 9.

Step 3. We analyzed the TQIM-H questionnaire's results via IPA. The results from the TQIM-H questionnaire were analyzed using the IPA methodology.

Step 4. We summarized the results from IPA. The results of an IPA graph were summarized and shown as refined TQIM-H factors.

### 2.3.3. Delphi Second Round

The second round of Delphi analyzed TQIM-H factors in the first and second quadrant from the IPA graph (from the first round) to design the TQIM-H conceptual framework. The process of the second round contains three steps:

Step 1. Experts were familiarized with the TQIM-H results analyzed by IPA. Expert panels analyzed the importance level and understand the factor's characteristics summarized from IPA analysis results (from the first round).

Step 2. We interviewed the experts to develop the TQIM-H framework. An in-depth interview with expert panels allowed the panel to introduce ideas useful for designing the TQIM-H conceptual framework. The ideas included an order of importance for each dimension, the relationship of each dimension/factor, and the structure of TQIM-H.

Step 3. We summarized the TQIM-H conceptual framework. Information from the experts was used to construct a summarized TQIM-H conceptual framework.

### 2.3.4. Delphi Third Round

The third round of Delphi allowed the experts to confirm the summarized TQIM-H conceptual framework from the second round. The process of the third round contains three steps:

Step 1. We interviewed the developed TQIM-H framework with expert panels. The summarized TQIM-H conceptual framework from the second round was reviewed and used for in-depth interviews with the experts from the first and second rounds.

Step 2. We re-analyzed with experts about TQIM-H conceptual framework. The author explained the TQIM-H conceptual framework and conclusions from the second round.

Step 3. We summarized the developed TQIM-H conceptual framework with experts. The expert panel evaluated and confirmed the summarized TQIM-H conceptual framework.

### 2.4. Technology Acceptance Model (TAM)

Technology Acceptance Model (TAM) questionnaire was used as a satisfaction measurement tool to evaluate the effective implementation of the TQIM-H conceptual framework to develop quality innovation projects in healthcare; ease of use, the comparison of the quality and innovation project development in healthcare through the TQIM-H conceptual framework, and the traditional developed innovation project in healthcare without the conceptual framework; and the practicality of the final TQIM-H conceptual framework.

Healthcare innovators or healthcare members who were related to healthcare innovation project development from the hospital in Southeast Asia participated in this study. They were selected based on their experience in developing healthcare innovations that allowed them to be able to evaluate the framework used in quality innovation development. The participants were familiarized with the developed TQIM-H conceptual framework. Then, they were asked via questionnaire for their opinions on the conceptual framework's ability to develop an effective innovation project. The process of TAM methodology contains four steps:

Step 1. We provided the TQIM-H framework and related information to participants. Fifty participants were familiarized with the TQIM-H conceptual framework thoroughly.

Step 2. The participants evaluated the conceptual framework using the TAM questionnaire. TAM questionnaire, which asked for the acceptance and efficacy of the TQIM-H when it was applied to real situations, was answered by the participants after they were familiarized with TQIM-H conceptual framework

Step 3. The participants evaluated the TQIM-H efficiency. All the tested constructs, except objective usability, were measured using 5-point Likert-type scales ranging from "strongly disagree" to "strongly agree".

Step 4. We analyzed the TAM results. TQIM-H conceptual framework acceptance and utilization to develop a healthcare innovation project were analyzed by descriptive statistics and expressed with means and standard deviation.

### 2.5. The Triangulation Technique with the Healthcare Innovation Development

To ensure the accuracy and validity of the results, triangulation was used to converge theories, methods, or observations to explain a phenomenon [33–35]. Triangulation also reduces biases that may have arisen from a single observation [36,37]. In this perspective, triangulation enables the confirmation of facts found in studies [38,39] by verifying the validity and reproducibility of the studies [40]. In this current study, the TQIM-H conceptual framework was presented to innovation developers so they could use the framework as a tool to develop quality innovation projects unique to challenges in each setting. The results yielded from developing projects were collected for verifying the TQIM-H conceptual framework. The process of the triangulation technique contains seven steps:

Step 1. The project team finding the pain point to develop the innovation. The innovation project development teams brainstormed to find the pain point in healthcare and then summarized to provide the healthcare innovation situation.

Step 2. The project team assigned the goal of the innovation development. Challenges in innovation development were defined and the Key Performance Index of the projects was set by the project teams.

Step 3. We provided the TQIM-H conceptual framework to develop healthcare innovation. The TQIM-H conceptual framework was proposed and presented to innovation project development teams so they were familiarized with the major characteristics of the TQIM-H conceptual framework and how to use it.

Step 4. The development team was familiarized with the TQIM-H concept. The project development teams studied the concept in detail, emphasizing the applicability TQIM-H conceptual framework.

Step 5. The development team developed the innovation following the TQIM-H framework. The innovation projects were initiated and the TQIM-H conceptual framework was used throughout the project.

Step 6. The authors monitored the project development. This was to ensure the adherence of project development to the TQIM-H framework.

Step 7. The efficiency of the innovation projects was evaluated using the Key Performance Index of the projects. The authors monitored the development of innovations that used the TQIM-H conceptual framework as a guideline for one year to evaluate the effect of using the framework on organizational performance. This in turn evaluated the effectiveness of TQIM-H.

### 3. Results

#### 3.1. A Literature Review

From a literature review, we concluded six key dimensions and 18 subfactors involving quality management in healthcare and five key dimensions and 17 subfactors involving innovation management in healthcare. The integration of these key factors and subfactors resulted in the total quality and innovation management in healthcare (TQIM-H) which consisted of seven key dimensions and 37 subfactors (Figure 3).

Table 2 demonstrates the details of TQIM-H, which has a key-factor-like characteristic; is used to develop innovative products or processes in healthcare under the framework of quality and leads to safe and standardized development. Column 1 shows seven TQIM-H dimensions including Context of the Environment (Internal and External), Leader, Planning, Support, Operation, Tools and Analysis method, and Improvement. Column 2 shows 37 TQIM-H subfactors which were derived from merging quality management in healthcare

factors (A), and innovation management in healthcare factors (B). Column 3 contains references to studies that yield each subfactor.

**Quality Management**

| Dimension | Sub-factor |
|---|---|
| A.1 Upper management | A1.1 Resources allocation from the leader<br>A1.2 Leader vision, Policy<br>A1.3 Assuming responsibility from the leader<br>A1.4 Supporting employees' suggestions from the leader |
| A.2 Customer-oriented | A2.1 Customer (patient etc.) satisfaction<br>A2.2 Solving the patient's complaints.<br>A2.3 An effective system for patient's rights |
| A.3 Ongoing improvement | A3.1 Quality audits<br>A3.2 Continuous solving problems<br>A3.3 Improving product and process quality<br>A3.4 Achieving quality standards |
| A.4 Employee engagement | A4.1 Informing the hospital's achievements<br>A4.2 Educating employees and training programs.<br>A4.3 Decision-making to solve problems |
| A.5 Operation management | A5.1 Organizational strategy<br>A5.2 Monitoring and evaluation<br>A5.3 Risk management<br>A5.4 Litigation law refers to the rules and practices |
| A.6 Tools and data analysis | A6.1 Information management<br>A6.2 Data integrity and security<br>A6.3 Data availability and accuracy |

**Innovation Management**

| Dimension | Sub-factor |
|---|---|
| B.1 Market needs | B1.1 Technological change<br>B1.2 Customer segment and customer needs |
| B.2 Organizational strategy | B2.1 Creating an organizational goal<br>B2.2 Alignment of innovation<br>B2.3 Innovation initiative with business needs and strategy |
| B.3 Organizational support | B3.1 Facilities e.g. laboratories, space, etc.<br>B3.2 Budgets<br>B3.3 Developing the educational center<br>B3.4 Human Resources |
| B.4 Administration system | B4.1 Process management<br>B4.2 Internal and External Networking<br>B4.3 Knowledge Management<br>B4.4 Portfolio Management |
| B.5 Best practice and excellence reputation | B5.1 Building distinctive competencies and competitive advantage<br>B5.2 Well-defined processes and formalized tools<br>B5.3 Establishing an innovation award<br>B5.4 Best practices documented and shared |

**Total Quality and Innovation Management in Healthcare**

| Dimension | Sub-factor |
|---|---|
| Context of the Environment (Internal & External) | A2.1 Customer (patient etc.) satisfaction<br>A2.2 Solving the patient's complaints.<br>A4.1 Informing the hospital's achievements<br>A5.4 Litigation law refers to the rules and practices<br>B1.1 Technological change<br>B1.2 Customer segment and customer needs |
| Leader | A1.1 Resources allocation from the leader<br>A1.2 Leader vision, Policy<br>A1.3 Assuming responsibility from the leader<br>A1.4 Supporting employees' suggestions from the leader |
| Planning | A5.1 Organizational strategy<br>B2.1 Creating an organizational goal<br>B2.2 Alignment of innovation<br>B2.3 Innovation initiative with business needs and strategy |
| Support | A4.2 Educating employees and training programs.<br>B3.1 Facilities e.g. laboratories, space, etc.<br>B3.2 Budgets<br>B3.3 Developing the educational center<br>B3.4 Human Resources |
| Operation | A2.3 An effective system for patient's rights<br>A4.3 Decision-making to solve problems.<br>A5.2 Monitoring and evaluation<br>A5.3 Risk management<br>B4.1 Process management<br>B4.2 Internal and External Networking<br>B4.3 Knowledge Management<br>B4.4 Portfolio Management<br>B5.1 Building distinctive competencies and competitive advantage<br>B5.3 Establishing an innovation award<br>B5.4 Best practices documented and shared |
| Tools and Analysis method | A6.1 Information management<br>B5.2 Well-defined processes and formalized tools<br>A6.2 Data integrity and security<br>A6.3 Data availability and accuracy |
| Improvement | A3.1 Quality audits<br>A3.2 Continuous solving problems<br>A3.3 Improving product and process quality<br>A3.4 Achieving quality standards |

**Figure 3.** The integration of TQIM-H. Code A: Quality management in healthcare factor, Code B: Innovation management in healthcare factor.

### 3.2. Case Study Analysis

The results from the analysis of 20 case studies that all proposed TQIM-H factors were used to develop all healthcare innovation projects as shown in Table 3. The most frequently used factors were in agreement with and impacted innovation development. We also found that TQIM-H factors were inclusive and sufficient for healthcare innovation development.

### 3.3. Delphi Study with Expert Panels

The results of the Delphi study were analyzed and presented in three parts.

#### 3.3.1. Delphi First Round

The TQIM-H, after the literature review and case study analysis, and before conducting the IPA analysis, had 37 factors. The Likert-scale questionnaire results of TQIM-H factors were analyzed using IPA methodology. After the IPA analysis, we retained 23 TQIM-H factors and called them refined TQIM-H factors.

Figure 4 shows the analysis of the importance and working performance level of the TQIM-H factors by the IPA graph. The *x*-axis provides the performance level of each TQIM-H factor scored by 30 healthcare experts while the *y*-axis shows the importance level of each TQIM-H factor provided by 30 expert panels. The TQIM-H factors in the first and second quadrant were used to develop the TQIM-H conceptual framework since they are important. This IPA analysis reduced TQIM-H factors from 37 factors to 23 factors. The descriptions for factors in the four quadrants (Figure 4) are demonstrated in Table 4.

**Table 2.** Total quality and innovation management in healthcare [41–62].

| Dimension | Sub-Factor | Reference |
|---|---|---|
| Context of the Environment (Internal and External) | A2.1 Customer (patient, etc.) satisfaction | [41,42,45,47,48,50–54,56,57,59,60,62] |
| | A2.2 Solving the patient's complaints. | [42,45,47,51–54,57,59,60,62] |
| | A4.1 Informing the hospital's achievements | [41,48,50,52,53,56,60–62] |
| | A5.4 Litigation law refers to the rules and practices | [41,44,46,48,51,53,55,56,61,62] |
| | B1.1 Technological change | [42,43,45–47,49,50,53,55,56,59–61] |
| | B1.2 Customer segment and customer needs | [41,45,47,51,52,54,56,59–61] |
| Leader | A1.1 Resources allocation from the leader | [41–53,55,56,58–60,62] |
| | A1.2 Leader vision, Policy | [41,43,45–48,50,52–58,60,61] |
| | A1.3 Assuming responsibility from the leader | [43,45,47,48,50,54,55,57,58,61,62] |
| | A1.4 Supporting employees' suggestions from the leader | [41,46–48,50,54,56,57,60,61] |
| Planning | A5.1 Organizational strategy | [42,45–47,49–54,56,57,60] |
| | B2.1 Creating an organizational goal | [42,43,46,50–52,56,57,59–61] |
| | B2.2 Alignment of innovation | [45,48–50,53,54,56,60] |
| | B2.3 Innovation initiative with business needs and strategy | [41,42,45,46,48,50,51,55,56,59,61,62] |
| Support | A4.2 Educating employees and training programs. | [43,46,51,54–57,59,61,62] |
| | B3.1 Facilities, e.g., laboratories, space, etc. | [41–43,46,49,50,52–54,56,57,60,61] |
| | B3.2 Budgets | [41,43,44,47,49,50,54,55,57–59,61,62] |
| | B3.3 Developing the educational center | [43,46,51,54,55] |
| | B3.4 Human Resources | [41–43,46,51,55,56,59,61] |
| Operation | A2.3 An effective system for patient's rights | [42,43,47,50,52,53,56,57,59,61,62] |
| | A4.3 Decision-making to solve problems. | [44,45,47,49,50,52,54–56,59,61] |
| | A5.2 Monitoring and evaluation | [41,42,45,47,48,51,52,54,57,61,62] |
| | A5.3 Risk management | [41,42,45–48,50,51,55,57,58,60–62] |
| | B4.1 Process management | [41–44,46,49–54,56,57,59–62] |
| | B4.2 Internal and External Networking | [41,44,47,48,51,53,56,61,62] |
| | B4.3 Knowledge Management | [41–43,45,47,49,50,52,54,55,57–59,61] |
| | B4.4 Portfolio Management | [43,51,54,56,61] |
| | B5.1 Building distinctive competencies and competitive advantage | [41,43,44,46–49,51,55,59,60,62] |
| | B5.3 Establishing an innovation award | [41,46,50,56,61] |
| | B5.4 Best practices documented and shared | [41,44,49,50,55,57,59,61,62] |
| Tools and Analysis method | A6.1 Information management | [42,45,46,49,50,52,56,59–61] |
| | B5.2 Well-defined processes and formalized tools | [41,44,46,47,52,55,57,60–62] |
| | A6.2 Data integrity and security | [41,43,46,47,50,52,53,55,57–59,61,62] |
| | A6.3 Data availability and accuracy | [41–43,45–48,50,51,53–55,57,60–62] |
| Improvement | A3.1 Quality audits | [41–43,45,47,48,50–53,56–59,61,62] |
| | A3.2 Continuous solving problems | [41–43,47,48,50–53,55–59,61,62] |
| | A3.3 Improving product and process quality | [41,42,47,50,51,53,55,56,58,59,61,62] |
| | A3.4 Achieving quality standards | [41,42,46,47,49,51,53,54,57,59,60] |

Code A: from quality management in the healthcare factor, Code B: from innovation management in the healthcare factor. Ref. [41]: (Fleiszer et al., 2015), Ref. [42]: (Lennox, Maher, and Reed, 2018), Ref. [43]: (Vergunst et al., 2020), Ref. [44]: (Ghannadpour, Zandieh, and Esmaeili, 2021), Ref. [45]: (Hussain, Ajmal, Gunasekaran, and Khan, 2018), Ref. [46]: (Moro Visconti and Morea, 2019), Ref. [47]: (Leite, Bateman, and Radnor, 2020), Ref. [48]: (Doyle et al., 2013), Ref. [49]: (Lennox, Linwood-Amor, Maher, and Reed, 2020), Ref. [50]: (de Fátima Castro, Mateus, and Bragança, 2015), Ref. [51]: (Asif, Searcy, Garvare, and Ahmad, 2011), Ref. [52]: (Kanji, 2005), Ref. [53]: (R. Chen, Lee, and Wang, 2020), Ref. [54]: (Hassini, Surti, and Searcy, 2012), Ref. [55]: (Bai, Dallasega, Orzes, and Sarkis, 2020), Ref. [56]: (Lopes, Scavarda, De Carvalho, Vaccaro, and Korzenowski, 2019), Ref. [57]: (Maynard et al., 2020), Ref. [58]: (Lindgreen, Antioco, Harness, and Van der Sloot, 2009), Ref. [59]: (AnAaker and Elf, 2014), Ref. [60]: (Santoyo-Castelazo and Azapagic, 2014), Ref. [61]: (Spangenberg, 2005), Ref. [62]: (Dervitsiotis, 2011).

From analyzing the TQIM-H factors, we found high importance levels but low performance levels in the first quadrant because all of these factors did not have the organizational regulation and KPI that measured tangible performance. Thus, healthcare workers did not give priority to improving and providing effective management. In addition, the organization did not have a policy and action plan on these factors. However, expert panels recommended that organizations should focus on and emphasize TQIM-H factors in the first quadrant since the factor in the first quadrant may be the key success factor in managing effective quality and innovation in healthcare. To achieve high organizational performance, hospitals should provide an organizational strategy and plan to efficiently manage these factors.

Then, after analyzing TQIM-H factors in the second quadrant, the author and expert panels found that these factors were important and had high performance because all of these factors were used as criteria and Key Performance Indicators (KPI) of an organization. Moreover, some factors in this quadrant represented medical regulation and quality stan-

dards; thus, the organization and healthcare workers paid attention to these factors and performed well. Therefore, such attributes must be maintained and exploited to achieve organizational maximum benefits as a potential competitive advantage. At this point, the factor in the second quadrant is important to sustain an optimum level of resources to suffice healthcare maximum benefits in Appendix A.

**Table 3.** TQIM-H factors analysis from 20 innovation case studies.

| Dimension | Sub-Factor | 1 | 2 | 3 | 4 | 5 | 6 | 7 | 8 | 9 | 10 | 11 | 12 | 13 | 14 | 15 | 16 | 17 | 18 | 19 | 20 | Sum |
|---|---|---|---|---|---|---|---|---|---|---|---|---|---|---|---|---|---|---|---|---|---|---|
| **Context of the Environment (Internal and External)** | A2.1 Customer (patient etc.) satisfaction | / | / | | / | | / | / | | / | | / | / | / | | / | / | | / | / | / | 14 |
| | A2.2 Solving the patient's complaints | | / | / | | | | | | / | | | | / | | / | | / | | | | 6 |
| | A4.1 Informing the hospital's achievements | / | | | / | | / | | | / | | / | | | / | | / | / | | / | / | 10 |
| | A5.4 Litigation law refers to the rules and practices | / | / | / | | / | / | | / | / | / | | / | | | / | / | / | / | / | / | 15 |
| | B1.1 Technological change | / | / | | / | / | | / | / | / | / | / | | / | / | | / | / | / | / | | 15 |
| | B1.2 Customer segment and customer needs | / | | / | / | | / | / | | / | | / | | | / | | / | / | / | | / | 11 |
| **Leader** | A1.1 Resources allocation from the leader | / | / | / | | / | / | | / | | / | | / | / | | / | | / | / | / | | 13 |
| | A1.2 Leader vision, Policy | / | | / | / | | / | / | / | | / | / | | / | | | / | / | / | | | 12 |
| | A1.3 Assuming responsibility from the leader | | / | | | / | | | / | / | | | / | | / | | | / | | / | / | 9 |
| | A1.4 Supporting employees' suggestions from the leader | | / | | | / | | / | | | | / | | | / | | / | | / | | | 7 |
| **Planning** | A5.1 Organizational strategy | / | / | | / | / | | / | / | / | | / | | / | / | / | / | | / | | / | 13 |
| | B2.1 Creating an organizational goal | | / | / | | | / | | | / | / | | | | / | | | / | | | | 7 |
| | B2.2 Alignment of innovation | / | | | / | | / | / | | / | | | / | / | | | / | | / | / | / | 11 |
| | B2.3 Innovation initiative with business needs and strategy | / | / | | | | / | | | | | / | | | / | | | | / | | | 6 |
| **Support** | A4.2 Educating employees and training programs. | | / | / | | / | / | / | | / | | | / | | / | / | | | / | | | 10 |
| | B3.1 Facilities, e.g., laboratories, space, etc. | / | | / | / | | / | | | / | | / | / | | / | | / | | / | / | / | 12 |
| | B3.2 Budgets | / | / | | / | | / | / | / | / | / | | / | / | / | | / | / | / | / | / | 16 |
| | B3.3 Developing the educational center | | | / | | | | | | | | | | / | | | | | / | | | 3 |
| | B3.4 Human Resources | / | / | / | | / | / | | / | / | | | / | | / | | / | / | | / | / | 13 |
| **Operation** | A2.3 An effective system for patient's rights | / | | / | / | | | / | | / | | | / | | | / | | | / | | | 8 |
| | A4.3 Decision-making to solve problems. | | / | | | | / | | / | | | | / | | / | | | / | | / | | 7 |
| | A5.2 Monitoring and evaluation | / | | / | / | | | / | | / | | / | | / | | / | / | / | | / | / | 12 |
| | A5.5 Risk management | / | / | | / | / | | / | | / | / | / | | / | | / | | / | / | / | / | 14 |
| | B4.1 Process management | / | / | / | / | / | / | / | / | / | / | / | / | / | / | / | / | / | / | / | / | 20 |
| | B4.2 Internal and External Networking | | / | / | | | / | | / | / | / | / | | / | | / | / | / | | / | / | 13 |
| | B4.3 Knowledge Management | / | / | | / | | / | / | | | | / | | / | / | | | | / | | / | 10 |
| | B4.4 Portfolio Management | / | | | | / | | | | | / | | | / | | | / | | | / | | 6 |
| | B5.1 Building distinctive competencies and competitive advantage | | / | | / | | / | / | / | | | | / | | / | | / | | / | / | | 11 |
| | B5.3 Establishing an innovation award | / | / | | / | / | / | | / | / | / | | / | / | | / | | | / | / | / | 15 |
| | B5.4 Best practices documented and shared | / | | / | | | / | | / | | / | | / | | | / | | / | | / | | 9 |
| **Tools and Analysis method** | A6.1 Information management | / | / | | / | / | | / | / | | / | / | / | | / | | / | | / | / | | 13 |
| | B5.2 Well-defined processes and formalized tools | / | / | / | | / | | / | | / | | / | | / | / | | / | | / | / | | 12 |
| | A6.2 Data integrity and security | / | | / | / | / | | / | / | / | | / | / | | / | / | / | / | / | / | / | 16 |
| | A6.3 Data availability and accuracy | / | / | / | / | / | / | / | | / | / | | / | / | / | | / | / | / | / | / | 17 |
| **Improvement** | A3.1 Quality audits | | / | | / | / | / | | / | / | | / | / | | / | | / | / | | / | / | 13 |
| | A3.2 Continuous solving problems | / | / | / | | / | | / | / | | / | | / | / | | / | / | | / | / | / | 14 |
| | A3.3 Improving product and process quality | | / | | / | | / | | | / | | / | | | / | | | / | | / | / | 9 |
| | A3.4 Achieving quality standards | / | | / | / | | / | / | / | / | / | / | | / | | / | / | | / | / | / | 15 |

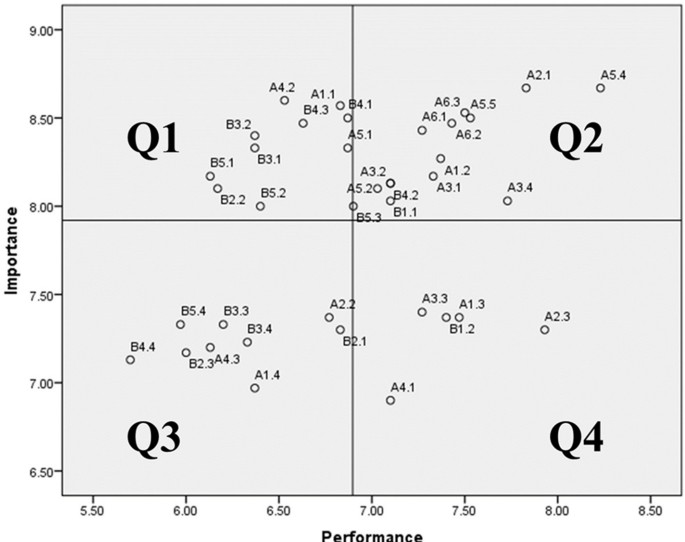

**Figure 4.** IPA of total quality and innovation management in the hospital.

**Table 4.** TQIM-H with IPA analysis.

| Quadrant | Characteristic | TQIM-H Factor |
|---|---|---|
| 1 | "Concentrate here" (high importance and low performance). This quadrant shows that a company's performance does not meet the importance level of its products and services. Therefore, management needs to focus on improving current products and services performance. | The first quadrant has 10 TQIM-H factors, including<br>- A1.1: Allocating resources<br>- A5.1: Organizational strategy<br>- B2.2: Alignment of innovation<br>- A4.2: Educating employees and training programs<br>- B3.1: Facilities, e.g., laboratories, space, etc.<br>- B3.2: Budgets<br>- B4.1: Process management<br>- B4.3: Knowledge Management<br>- B5.1: Building distinctive competencies and competitive advantage<br>- B5.2: Well-defined processes and formalized tools. |
| 2 | Quadrant 2: "Keep up the good work" presented high importance and high performance of each TQIM-H factor. Attributes plotted in this area show that the hospital must continue to perform well, as the attributes are considered important. The attributes in this quadrant may be viewed as a set of opportunities to continue doing a good job over competitors. | The second quadrant has 14 TQIM-H factors, including<br>- A2.1: Customer (patient, etc.) satisfaction<br>- A5.4: Litigation law refers to the rules and practices<br>- B1.1: Technological change<br>- A1.2: Leader vision, Policy<br>- A5.2: Monitoring and evaluation<br>- A5.3: Risk management<br>- B4.2: Internal and External Networking<br>- B5.3: Establishing an innovation award<br>- A6.1: Information management<br>- A6.2: Data integrity and security<br>- A6.3: Data availability and accuracy<br>- A3.1: Quality audits<br>- A3.2: Continuous solving<br>- A3.4: Achieving quality standards. |
| 3 | Quadrant 3: "Low priority" (low importance and low performance). Customers perceive attributes in this area as unimportant and communicate that the company is not performing well. | The third quadrant has nine TQIM-H factors, including<br>- A2.2 Solving the patient's complaints<br>- A1.4 supporting employees' suggestion<br>- B2.1 Creating an organizational goal<br>- B2.3 Innovation initiative with business needs and strategy<br>- B3.3 Developing the educational center<br>- B3.4 Human Resources<br>- A4.3 Decision-making to solve problems<br>- B4.4 Portfolio Management, and<br>- B5.4 Best practices documented and shared. |
| 4 | Quadrant 4: "Possible overkill" (low importance and high performance). For each attribute in this area, customers evaluate its performance as exceeding its importance. Therefore, too much attention paid to this area could represent overkill concerning the use of resources that could be better directed to other areas, although high performance on an attribute in this area could be considered a strength in that it may enable the company to attract new customers (Gates and Amarani, 1992). | The fourth quadrant has five TQIM-H factors, including<br>- A4.1 Informing the hospital's achievements<br>- B1.2 Customer segment, and customer needs<br>- A1.3 Assuming responsibility<br>- A2.3 An effective system for patient's rights<br>- A3.3 Improving product and process quality. |

After brainstorming and analysis among the author and expert panels, the attribute situated in quadrant three and quadrant four had low importance. The attribute was successfully performed but was unfortunately deemed irrelevant to the management of quality and innovation in healthcare. As such, there is no need for any changes in the efforts or resources allocated. On the other hand, perhaps it is more beneficial to curtail the resource allocation and redeploy the efforts to the other attribute that needs immediate action. Thus, in this study, fourteen TQIM-H subfactors in quadrant three and quadrant four were omitted. Therefore, the number of TQIM-H subfactors was reduced from 37 subfactors to 23 factors from the first and second quadrants. These 23 subfactors were used in the next round.

### 3.3.2. Delphi Second Round

All experts agreed that the 23 sub-factors (in seven dimensions) derived from the first and second quadrants were the key factors in managing healthcare quality and innovation framework. We found that the experts presented their ideas on how to develop the TQIM-H conceptual framework similarly. All experts agreed on the order of the framework that should be considered according to their importance as the following. First, the Context of the Environment (Internal and External) was important and should be investigated before developing other processes to allow the comprehension of problems, customer needs, and changes. Second, Leaders should lead, plan, and drive innovation development. The third dimension that should be focused on is Planning, which was used to set the direction of working processes in developing innovation toward the policy. Then, Operate referred to processes in developing innovations according to the plan. Operate consisted of Process management, Risk management, Knowledge management, etc. Tools and Analysis methods were required for solving problems. These concepts should be used and improved continuously in the innovation developing process and must be supported by all stakeholders in the organizations to allow effective innovation in healthcare. Data from the operation were then analyzed and monitored by the Tools and Analysis method. Moreover, Supporting the development system was also important. Supporting consisted of Budgets, Employee education, and Facilities. The last but important step to be considered was Improvement. Developing innovation was effective and sustainable when development occurred continuously. Therefore, Improvement was a foundation and background for every dimension.

### 3.3.3. Delphi Third Round

The TQIM-H conceptual framework developed in the previous round was confirmed by the healthcare experts. All healthcare experts agreed that the developed framework was effective for developing healthcare innovation projects. Additionally, they all agreed that the developed framework provided proper coverage; was completed for developing innovative projects; and agreed with the healthcare quality framework. The final conceptual framework that would lead to effective innovation in healthcare consisted of seven dimensions. For healthcare innovators who would like to develop innovation, the full-cycle implementation of the TQIM-H conceptual framework in the projects would lead to the efficient and systematic development of innovation in healthcare.

The proposed TQIM-H yielded after the literature review, before conducting the IPA analysis, had 37 factors. The Likert-scale questionnaire results of TQIM-H factors were analyzed using IPA methodology. After the IPA analysis, we retained 23 TQIM-H factors and called them refined TQIM-H factors.

After finishing three rounds of Delphi study with healthcare experts, the TQIM-H factors were developed as a conceptual framework that could be used as a guideline for developing innovation in hospitals and that agreed with the healthcare quality framework. This TQIM-H conceptual framework still had seven dimensions but contained fewer subfactors since 14 nonsignificant or unrelated subfactors were omitted. The remaining 23 subfactors were used as a key management factor of the TQIM-H conceptual framework.

*3.4. TAM*

After understanding the TQIM-H conceptual framework, the feasibility and acceptability of the TQIM-H conceptual framework were evaluated based on the TAM. The participant information is shown in Table 5. The questionnaires which consisted of items shown in Table 6 were sent to 50 healthcare innovators and healthcare quality staff to obtain their comments on the effectiveness, coverage, completeness, and applicability of the TQIM-H conceptual framework. The results showed that TQIM-H conceptual framework decreased time wasted, provided an effective process for developing quality and innovation projects in healthcare, and completely covered the development of quality and innovation projects in healthcare.

**Table 5.** General data of the sample group.

| Respondents' Demographics | Frequency | Percent |
|---|---|---|
| Gender | | |
| Men | 19 | 38 |
| Women | 31 | 62 |
| Total | 50 | 100 |
| Age | | |
| <30 years | 3 | 6 |
| 30–39 years | 11 | 22 |
| 40–49 years | 19 | 38 |
| 50–59 years | 12 | 24 |
| >60 years | 5 | 10 |
| Total | 50 | 100 |
| Level of education | | |
| Bachelor's degree | 28 | 56 |
| Master's degree | 16 | 32 |
| Doctorate | 6 | 12 |
| Total | 50 | 100 |
| Position | | |
| President/Director/Manager | 17 | 34 |
| Physician/Dentist/Pharmacist | 12 | 24 |
| Medical technician/Radiologist/Physiotherapist/Nutritionist | 2 | 4 |
| Nurse/Nursing Assistant | 8 | 16 |
| Customer service | 2 | 4 |
| Office workers/Support staff | 7 | 14 |
| Other | 2 | 4 |
| Total | 50 | 100 |
| Working Experience | | |
| <10 years | 8 | 16 |
| 10–20 years | 32 | 64 |
| >20 years | 10 | 20 |
| Total | 50 | 100 |
| The TQIM-H conceptual framework experience | | |
| Not used to | 50 | 100 |
| Used to | 0 | 0 |
| Total | 50 | 100 |
| Preference to use the TQIM-H conceptual framework | | |
| Acceptation | 50 | 100 |
| Rejection | 0 | 0 |
| Total | 50 | 100 |

*3.5. Triangulation with the Healthcare Innovation Development*

The TQIM-H conceptual framework was tested and validated by being used to develop a healthcare innovation project from the beginning until finished in a selected hospital. The hospital is a large hospital in Thailand that is accepted in the Southeast Asia region as a prototype hospital. Moreover, the case study hospital is also accredited by Joint Commission International (JCI) certification. Therefore, the applicability that was confirmed in this hospital also assured the applicability in other regional hospitals.

**Table 6.** The technology acceptance model with the TQIM-H conceptual framework.

| The Program Characteristic | Mean | SD |
|---|---|---|
| 1. Effective implementation of TQIM-H conceptual framework to develop quality innovation projects in healthcare | 4.67 | 0.52 |
| (1.1) Decreases time wasted in developing quality and innovation projects in healthcare | 4.75 | 0.65 |
| (1.2) Provides an effective process for developing quality and innovation projects in the healthcare | 4.62 | 0.54 |
| (1.3) Be comprehensive and completely cover the development of quality and innovation projects in the healthcare | 4.74 | 0.69 |
| (1.4) Is a modern and acceptable conceptual framework | 4.56 | 0.78 |
| 2. Ease of use | 4.61 | 0.92 |
| (2.1) The objective of using the TQIM-H conceptual framework is clear | 4.55 | 0.80 |
| (2.2) The operation procedure of the TQIM-H conceptual framework is clear and easy to understand | 4.64 | 0.48 |
| (2.3) The conceptual framework is easy to learn and understand. Self-study using the instructions TQIM-H conceptual framework is easy | 4.58 | 0.32 |
| (2.4) A healthcare innovator can easily use the TQIM-H conceptual framework to develop quality and innovation projects in the healthcare | 4.76 | 0.58 |
| (2.6) TQIM-H conceptual framework is easy to use. | 4.54 | 0.65 |
| 3. User Interface | 4.60 | 0.49 |
| (3.1) TQIM-H conceptual framework is attractive | 4.68 | 0.75 |
| (3.2) TQIM-H conceptual framework is up-to-date | 4.60 | 0.92 |
| (3.3) The diagram of the TQIM-H conceptual framework is appropriate | 4.52 | 0.81 |
| 4. The comparison of the quality and innovation project development in healthcare through the TQIM-H conceptual framework and the traditional developed innovation project in healthcare without the conceptual framework. | N/A | N/A |
| (4.1) The conceptual framework reduces time spent collecting, analyzing, and processing to develop quality and innovative projects in healthcare | N/A | N/A |
|     Before the TQIM-H conceptual framework is used | 3.34 | 0.67 |
|     After the TQIM-H conceptual framework is used | 4.54 | 0.83 |
| (4.2) The conceptual framework reduces skills, expertise and reduces decisions using experience to measure and evaluate develop quality and innovation projects in healthcare | N/A | N/A |
|     Before the TQIM-H conceptual framework is used | 3.48 | 0.59 |
|     After the TQIM-H conceptual framework is used | 4.76 | 0.68 |
| (4.3) The conceptual framework provides a systematic work process that is clear so using the program is convenient and easy. | N/A | N/A |
|     Before the TQIM-H conceptual framework is used | 3.12 | 0.95 |
|     After the TQIM-H conceptual framework is used | 4.82 | 0.87 |
| (4.4) The conceptual framework reduces work processes and eliminates the duplication of operations. | N/A | N/A |
|     Before the TQIM-H conceptual framework is used | 3.26 | 0.75 |
|     After the TQIM-H conceptual framework is used | 4.86 | 0.64 |
| 5. The practical concept of the TQIM-H conceptual framework | 4.66 | 0.38 |
| (5.1) TQIM-H conceptual framework can be applied to quality and innovation project development in healthcare effectively. | 4.70 | 0.96 |
| (5.2) TQIM-H conceptual framework leads to the improvement of processes involved in the development of quality and innovation projects in healthcare. | 4.62 | 0.94 |

N/A: not applicable.

Healthcare Innovation Situation

This project was initiated to solve the ineffective drug management in the inpatient department and complaints from medication errors. Nurses were responsible for preparing unit dose medications for each inpatient. Increased workload and the lack of a verification process led to missing doses and dispensing them to the wrong patients. Additionally, medications require different storage temperatures, so transferring medications to each storage point required multiple temperature-controlled storage boxes. This led to the idea of creating a medication cart that was able to (1) store unit dose medications, which were

ready to deliver to each patient, (2) allow the separation of medication according to the time taken, (3) control temperature to be at different temperatures that were optimal for different medications, (4) allow the verification of patients' name to reduce errors, and (5) reduce IPD nurses' workload. The innovation project was then developed using the TQIM-H conceptual framework as described in Table 7.

**Table 7.** The external dataset used for triangulation with TQIM-H conceptual framework.

| TQIM-H | Procedure |
|---|---|
| Context of the Environment (Internal and External) | - The working team collected issues and complaints from IPD services.<br>- The working team analyzed the root cause of the problems with users to design innovative products that served the user's needs.<br>- The working team analyzed the IPD working process to search for statements and pain points.<br>- The working team consulted with material experts to design the medication cart as an innovative product. |
| Leader | - Leaders saw the importance of problems which were medication errors occurring in IPD.<br>- Leaders set the policy to develop innovation to prevent medication errors.<br>- Leaders appointed teams who worked on developing the innovation.<br>- Leaders monitored and allocated resources necessary for innovation development. |
| Planning | - A working plan was created.<br>- The plan to develop innovation must be aligned to solutions for the problems, medication errors.<br>- The plan to develop innovation must be aligned with the technological trends involved in IPD services. |
| Operation | - The working team healthcare staff involving IPD services brainstormed to acquire in-depth details, customer journey, and pain points.<br>- The working team designed the innovation with healthcare staff involved in IPD care.<br>- The working team consulted with material experts to plan the assembly of the medication cart<br>- To prove the concept, a prototype IPD medication cart was constructed and experimented with in the IPD.<br>- The working team monitored to allow continuous improvement of the IPD cart.<br>- The innovation project was furnished according to the final concept. |
| Tools and Analysis method | - Statistic tools were used to analyze the project information<br>- Material tools were used to investigate the most optimal material for the IPD cart. |
| Support | - Training programs on the design and development of product innovation were provided to employees.<br>- Budgets were allocated to the IPD medication cart project.<br>- Information on characteristics of IPD medications and services was provided to the team conducting the project.<br>- Materials, technologies, and facilities, e.g., IPD wards were allocated to the project. |
| Improvement | - Healthcare quality committees were appointed to evaluate, audit, and control quality and measure the risk of the IPD medication cart following organizational quality standards before launch.<br>- Quality audits were used to evaluate the IPD medication cart during use.<br>- Usability and ability to solve the problems of the IPD cart were monitored.<br>- The cart (product) and working process were continuously improved to maximize efficiency. |
| Organizational performance measurement | - Data from 6-month monitoring of the IPD medication cart were developed following the TQIM-H conceptual framework. We found that<br>- IPD patients' satisfaction was increased. Most of the patients received medications correctly and on time. The service and innovation were credible.<br>- The IPD medication cart was able to stably maintain the target temperatures.<br>- Medication error occurrence was reduced to zero.<br>- The frequency of complaints was decreased by approximately 80%.<br><br>IPD service time was reduced by 48%.<br>Nurse man-hours were reduced by 33%. |

Information from the project confirmed that the TQIM-H conceptual framework helps develop innovation systematically since the framework showed key success factors, orderly processes, and details in each step of innovation development. This allowed the developer to follow the framework step-by-step, thus increasing the chance of success and reducing waste caused by errors.

Anyway, to assure the reliability of the developed conceptual framework for developing healthcare innovation, in future work, this TQIM-H conceptual framework should be further used as a guideline for developing a variety of healthcare innovations, e.g., product

innovation, process innovation, and business model innovation in hospitals in several areas in Southeast Asia. This is to confirm the applicability of and to continuously improve the framework.

*3.6. The Refined TQIM-H Conceptual Framework*

The refined TQIM-H conceptual framework consisted of seven dimensions that facilitated effective innovation in healthcare (Figure 5).

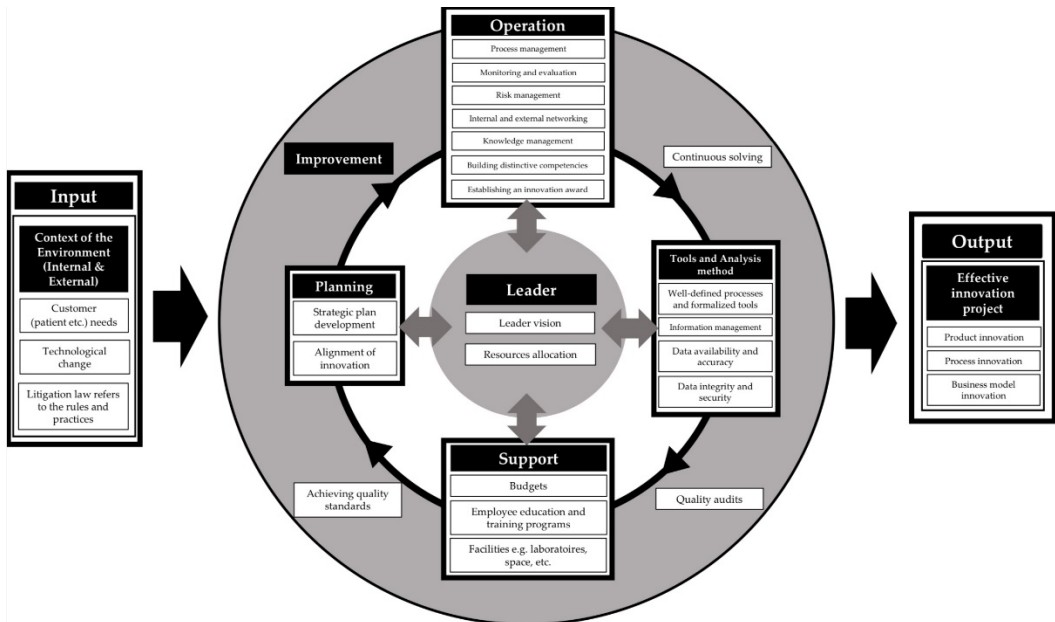

**Figure 5.** The conceptual framework of TQIM-H.

The TQIM-H conceptual framework is a structural management framework whose key dimensions and subfactor components could be used as a guideline for the development of innovation projects in healthcare and still allow the adherence to the quality principle in hospitals. The details of this framework are listed below.

Context of the Environment (Internal and External) dimension was the first part to be considered. It aimed to investigate problem statements and analyze pain points. In-depth analysis of factors from inside and outside of the organization led to the realization of patient needs and innovation characteristics, which in turn increase healthcare users' satisfaction and acceptance. In healthcare, Context of the Environment was related to changes in technologies involved in pharmaceuticals and therapeutics, and changes in diseases. In addition, healthcare was tightly regulated by medical ethics, rules, and laws. Therefore, innovation projects developed in hospitals must comply with these regulations. Then, when the problems were defined, Leader became important since leaders could set the policy and allocate resources important for supporting the innovation. Since healthcare personnel, e.g., physicians, pharmacists, nurses, customer service, etc., consist of a variety of professions or occupations with diverse backgrounds, they have diverse working cultures and working aspects. Therefore, Leader was required as a center of communication and a director of projects so that different people could work together harmoniously. The Leader acted centrally to drive and control the development of innovation in every stage, which allowed smooth and continuous development processes. To effectively develop the projects and organizations, the Leader must be accepted by all stakeholders, set valid visions, delegate, and distribute works to suitable individuals, and holistically oversee the projects or organizations. The third part to be considered and developed was Planning, which aimed to design and plan the development process from the beginning to the end (end-to-end). This allowed a clear goal for each stage of development. Planning was important for every

working process, especially in work that involves high-stake processes. Healthcare is also included as such since it involves human life. Therefore, planning must be executed before any action. Additionally, monitoring the healthcare development, predicting obstacles, and planning the solutions were easier with Planning. After Planning, Operation became important. Operation consisted of Process management that allows the initiation of the development process. For effective innovation development in hospitals, alignment with the quality framework required several tools including innovation process development tools and quality process development tools. Moreover, Monitoring and evaluation was another significant process that required experts and criteria for evaluating the effectiveness, adherence to medical principles, and risks of developed innovation projects. Next, Risk management included evaluation and mitigation of the risks. Since hospitals aimed to minimize and mitigate patient hazards, Risk management was important as prevention and solution measures. Last, Knowledge management was required since healthcare is multidisciplinary. Knowledge centralization and distribution were required to allow smooth operation and share understanding in innovation development in healthcare. To allow the effective Operation to occur, Tools and Analysis methods, including data management and analysis, were important for the analysis of the scenario to provide directions for the development. Important Tools and methodology consisted of quality tools and innovation tools. Innovation tools stimulated innovative thinking, which would facilitate innovation development in hospital while Quality tools controlled and assured the adherence of the working process to healthcare standards, and minimized risks. Furthermore, Support, manifesting as Budgets, Employee education, and Facilities, was also important in developing innovation. Since healthcare works were, in general, specialized and routine, knowledge and facility support stimulated innovation development in hospitals. The last part was the Improvement dimension. Continuous improvement was needed in healthcare since changes are normal in healthcare. Emerging diseases, novel therapeutic options, and changes in health lifestyle force the healthcare industry to continuously develop. Thus, continuous improvement is important and should be the foundation or philosophy of healthcare management since it maximizes working capabilities and keeps the working methods up-to-date.

## 4. Discussion and Implications

This study used multiple methodologies including a literature review, multiple case studies, a Delphi study with healthcare experts, and the TAM technique to create the TQIM-H conceptual framework. The validity of the framework was tested with an external dataset. The key components of the final TQIM-H conceptual framework could be classified into seven dimensions or subfactors. Since TQIM-H is a unique tool used to integrate quality and innovation management, the direct comparison of TQIM-H to other tools with the same purpose is not possible. However, the TQIM-H framework developed in this study was completed and systematically explained the relationship and usage of each key success factor in managing healthcare innovation projects.

The relationship between quality management and innovation management in industries was not conclusive. In addition, studies addressing the relationship between two management philosophies in the healthcare industry are scarce [63–65]. To the best of our knowledge, there is only one study, from Tonjang and Thawesaengskulthai (2020) [30], that addressed such a relationship and found that the relationship was positive. The study showed that innovation maximizes the capability of quality management and clarifies the goal of quality management. Meanwhile, quality directs the direction of innovation development so the development adheres to medical safety standards and medical law. Ultimately, these two management frameworks have the same goal, i.e., to serve customer needs. Nevertheless, the integration of quality and innovation management in the hospital context has never been conducted. Therefore, there was no platform for innovation development so innovation development highly depends on experiences of the healthcare staff who understand effective innovation management that does not conflict with the

healthcare quality framework [66]. Normally, healthcare professionals lack experience in management [67,68]. Therefore, the development of the integrated framework of quality management and innovation management, containing key factors, the relationship among dimensions, and systematic usage of dimensions, in this study allows healthcare professionals to compensate for the lack of experience so the chance of successful development of innovations is enhanced.

During the development of the TQIM-H conceptual framework, we found that Context of the Environment, an input of necessary information to the system, was the first component that should be focused on. Other management frameworks that were applied in healthcare organizations, e.g., TQM, risk management, and knowledge management also pointed out that data input is the most important beginning step in management [69]. We also illustrated the roles of Leader in innovation development. Dargan and Shucksmith (2008) [70] found that not only the project team should understand the important issues of the problems but leaders should also understand the issues. Research has demonstrated the importance of leadership engagement and showed that projects that receive leaders' attention and support are more likely to be successful than inattention projects [71]. Since healthcare usually has diverse departments or sections, the top-down operational process is more effective than the bottom-up process. Moreover, several studies also supported that clear plans and systematic provision of analysis tools increased the effectiveness in managing and developing projects, and reduced failure chances [72–74]. Planning is also important in healthcare since it involves human life. Moreover, Leede, Looise, and Alders (2002) [75] proposed that developing innovative projects for effective changes must base all managerial dimensions on continuous improvement to allow the maximization of all processes since healthcare deals with emerging diseases, novel medical technologies, and dynamic health behaviors.

The structure and details of the TQIM-H conceptual framework that was developed in this study are suitable for the development of healthcare innovation that agrees with the quality framework since the dimensions and subfactors contained in the TQIM-H conceptual framework came from the integration of innovation management and quality management. Furthermore, the structure was similar to the structure of the healthcare management system in Southeast Asia, e.g., Development was initiated from needs and technological changes while law and medical ethics strictly complied; Leaders were the center of top-down processes; Plans were clearly designed before the execution; Operation was conducted as the planning process to minimize risk to patients; and Continual improvement was conducted to deal with changes. In addition, there were details indicating healthcare organization culture, e.g., patient-centered and multidisciplinary. Additionally, this study used healthcare experts in the process of integrating quality management and innovation management. This conceptual framework that was developed according to these healthcare backgrounds allowed healthcare innovators to use this framework for developing innovative projects in hospitals effectively and easily.

This study has several strengths. First of all, we used mixed methods and triangulation to ensure that our data were completed and sufficient for developing the TQIM-H conceptual framework. Internal validity was increased and biases were reduced by such practice. Second, the number of case studies in this research was large. Third, the TQIM-H conceptual framework was developed by a three-round Delphi study with healthcare experts, ensuring that personal biases were reduced. Lastly, the triangulation that incorporates the real-life innovation project in healthcare to confirm the practicality of the TQIM-H conceptual framework also illustrates the use of the framework in a real situation.

Anyhow, there were some limitations in this study. TQIM-H is an integrated framework that was developed for managing quality innovation in healthcare using healthcare experts in Southeast Asia. The experience of the experts is limited to the regulations in Southeast Asia. Additionally, the efficiency of the framework was evaluated by using an innovation case study in Southeast Asia. Therefore, the TQIM-H framework agrees with healthcare management in Southeast Asia. The application of the framework to other

regions or industries may be limited and needs future studies to confirm the generalizability of the framework. At the individual project level, applying the TQIM-H framework to settings outside Southeast Asia or other industries demands the adjustment of the TQIM-H framework. In addition, the TQIM-H conceptual framework shows a big picture of the key management dimensions used as a guideline for managing healthcare innovation by complying with the quality framework. Details of each dimension or subfactor, e.g., process, procedure, measurement, controlled tools, etc. are still lacking and should be investigated in future research. We suggest that future research should include (1) detailing the dimensions and/or subfactors of TQIM-H so the user can follow the framework easier, and (2) applying the TQIM-H conceptual framework for the development of more diverse healthcare innovations to validate and confirm its generalizability.

## 5. Conclusions

The success of a healthcare innovation project depends on effective management and alignment with the healthcare quality context. In a previous study, we found a positive relationship between quality management in healthcare and innovation management in healthcare. However, there has never been an integration of the two management frameworks. This study developed a new integrated framework of TQIM-H from four methodologies. The new conceptual framework consisted of seven dimensions including Context of the Environment (Internal and External), Leader, Planning, Support, Operation, Tools and Analysis method, and Improvement. Verification of the framework by an IPD medication cart project confirmed that the TQIM-H framework was useful in developing effective healthcare innovation.

**Author Contributions:** Conceptualization, S.T. and N.T.; methodology, S.T. and N.T.; investigation, S.T. and N.T.; writing—original draft preparation, S.T.; writing—review and editing, S.T. and N.T.; supervision, N.T.; project administration, S.T. and N.T. All authors have read and agreed to the published version of the manuscript.

**Funding:** This research was funded by the 100th Anniversary Chulalongkorn University Fund for Doctoral Scholarship from Chulalongkorn University, Bangkok, Thailand.

**Institutional Review Board Statement:** Ethics approval was not required in this research.

**Informed Consent Statement:** All participants consented to participate in this study. They were given the opportunity to ask questions regarding the study and the collection of the assessment scores. They received answers to any questions regarding the study.

**Data Availability Statement:** Not applicable.

**Conflicts of Interest:** The authors declare no conflict of interest.

## Appendix A

**Table A1.** IPA Analysis Results of TQIM-H.

| TQIM-H | Importance Level | | Performance Level | | Quadrant |
|---|---|---|---|---|---|
| | Mean | SD | Mean | SD | |
| A2.1 Customer (patient, etc.) satisfaction | 8.67 | 0.55 | 7.83 | 0.91 | Q2 |
| A2.2 Solving the patient's complaints. | 7.37 | 1.27 | 6.77 | 0.86 | Q3 |
| A4.1 Informing the hospital's achievements | 6.9 | 1.58 | 7.1 | 1.03 | Q4 |
| A5.4 Litigation law refers to the rules and practices | 8.67 | 0.71 | 8.23 | 1.04 | Q2 |
| B1.1 Technological change | 8.03 | 0.76 | 7.1 | 1.47 | Q2 |
| B1.2 Customer segment and customer needs | 7.37 | 1.03 | 7.4 | 0.97 | Q4 |
| A1.1 Allocating resources. | 8.57 | 0.63 | 6.83 | 1.02 | Q1 |
| A1.2 Leader vision, Policy | 8.27 | 0.69 | 7.37 | 1.07 | Q2 |
| A1.3 Assuming responsibility. | 7.37 | 1.22 | 7.47 | 1.28 | Q4 |
| A1.4 Supporting employees' suggestion | 6.97 | 1.4 | 6.37 | 1.22 | Q3 |
| A5.1 Organizational strategy | 8.33 | 0.88 | 6.87 | 1.17 | Q1 |
| B2.1 Creating an organizational goal | 7.3 | 1.02 | 6.83 | 0.87 | Q3 |
| B2.2 Alignment of innovation | 8.1 | 0.8 | 6.17 | 0.91 | Q1 |
| B2.3 Innovation initiative with business needs and strategy | 7.17 | 1.42 | 6 | 0.95 | Q3 |
| A4.2 Educating employees and training programs | 8.6 | 0.56 | 6.53 | 1.2 | Q1 |
| B3.1 Facilities, e.g., laboratories, space, etc. | 8.33 | 0.76 | 6.37 | 1.35 | Q1 |
| B3.2 Budgets | 8.4 | 0.86 | 6.37 | 1.25 | Q1 |
| B3.3 Developing the educational center | 7.33 | 0.96 | 6.2 | 0.89 | Q3 |
| B3.4 Human Resources | 7.23 | 1.01 | 6.33 | 1.15 | Q3 |
| A2.3 An effective system for patient's rights | 7.3 | 1.06 | 7.93 | 1.14 | Q4 |
| A4.3 Decision-making to solve problems. | 7.2 | 1.58 | 6.13 | 1.25 | Q3 |
| A5.2 Monitoring and evaluation | 8.1 | 0.71 | 7.03 | 1.19 | Q2 |
| A5.5 Risk management | 8.5 | 0.63 | 7.53 | 0.9 | Q2 |
| B4.1 Process management | 8.5 | 0.51 | 6.87 | 0.97 | Q1 |
| B4.2 Internal and External Networking | 8.13 | 0.82 | 7.1 | 1.37 | Q2 |
| B4.3 Knowledge Management | 8.47 | 0.57 | 6.63 | 1.3 | Q1 |
| B4.4 Portfolio Management | 7.13 | 1.01 | 5.7 | 1.21 | Q3 |
| B5.1 Building distinctive competencies and competitive advantage | 8.17 | 0.75 | 6.13 | 1.43 | Q1 |
| B5.3 Establishing an innovation award | 8 | 0.74 | 6.9 | 1.16 | Q2 |
| B5.4 Best practices documented and shared | 7.33 | 1.03 | 5.97 | 1.1 | Q3 |
| A6.1 Information management | 8.43 | 0.77 | 7.27 | 1.11 | Q2 |
| B5.2 Well-defined processes and formalized tools | 8 | 0.95 | 6.4 | 1.04 | Q1 |
| A6.2 Data integrity and security | 8.47 | 0.82 | 7.43 | 1.5 | Q2 |
| A6.3 Data availability and accuracy | 8.53 | 0.78 | 7.5 | 1.04 | Q2 |
| A3.1 Quality audits | 8.17 | 0.83 | 7.33 | 1.06 | Q2 |
| A3.2 Continuous solving | 8.13 | 0.73 | 7.1 | 0.84 | Q2 |
| A3.3 Improving product and process quality | 7.4 | 0.81 | 7.27 | 1.14 | Q4 |
| A3.4 Achieving quality standards | 8.03 | 0.89 | 7.73 | 1.08 | Q2 |

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
