# Peer review of "Total Quality and Innovation Management in Healthcare (TQIM-H) for an Effective Innovation Development: A Conceptual Framework and Exploratory Study"

_asi, doi:10.3390/asi5040070_

Round 1

Reviewer 1 Report

The manuscript doesn't add new informations to international literature

Reviewer 2 Report

The study is on total quality and innovation management in healthcare (TQIM-H) aimed for an effective innovation development in an effort to respond to customer needs and changing world. In my view the study has social value and is of interest to some readers in healthcare.

The authors however, need to improve on some of the issues indicated below:

1. Abstract: The abstract needs to clearly articulate and unpack the issue/problem that is bedeviling the field of healthcare in relation to the current study topic. This will make the gap in the study clear and it allows the readers to see where the present study takes off from. The major contributions of the study must also be specified or, if specified, the authors must indicate whether this is a novel contribution to the field of study.

Some these issues raised here are only clear in the concluding remarks.

2. The introduction can be improved by adding information on the contribution of the study at the end of the section or as a subsection under introduction or after the literature section. 

3. The conclusion section must end with areas for future research directions related to this study. A research study of this nature cannot be exhaustive. Therefore, a paragraph or so on areas for further research must be added. 

4. While I can confirm that the delimitations of the study are provided under the discussion section, limitations of the study may also be included in an appropriate section. 

5. The Reference section needs some thorough revision for consistency. For example, some journal names are start with lowercase letters which others are correctly written, e.g. 

Incorrect example:

Ref 1: International journal of environmental research and public health

Correct example:

Ref 2: Total Quality Management & Business Excellence

I suggest the authors check and correct this for the whole reference list. 

Reviewer 3 Report

I reviewed the paper you sent, but had to read it again and again because it does not seem to flow logically (and that might be because English is obviously not their first language). I am having the following comments and suggestions to make it more readable and improve the quality of the paper:

  1. Pages 2-4 seem to describe the 7 TQIM-H dimensions and their associated 38 subfactors.  However, the actual TQIM-H conceptual framework illustrated on page 19 does not include all of the subfactors listed on pages 2-4.  That might be because they eliminated some of them as a result of their study, but that’s not clear.  Then on page 4 we start referring to the conceptual framework again, but we have no idea what the framework is at this point.  It is recommended that they introduce the conceptual framework first, the launch into the process of developing it. That way, the paper will provide a logical context to the study and the research approach will be easier to follow.
  2. Page 19 illustrates the final conceptual framework, which seems to list all of the “ingredients” they believe are part of a successful innovation in healthcare approach.  The bulk of the paper is essentially made up of how they determined these “ingredients” – but I don’t see (and perhaps I’m mistaken) how quality is measured/controlled in this approach, what are the essential elements/processes of quality in an health-focused innovation exercise, etc., which I thought was the purpose of this paper (as defined in lines 38-40 on page 1).  Or – maybe there is a narrower definition of quality that they are looking at, in which case that must be explained up front.
  3. In line 175 on page 7, it seems to refer that only 1 hospital (“the hospital”) participating in this study.  This is a point of clarification only because in reading the rest of the paper it seems as though more hospitals participated, but the paper needs to be consistent.  If only 1 hospital participated, however, is it fair to say that the test case proving the TQIM-H methodology works.
  4. The TQIM-H core elements do not include the traditional ideation sessions, focus on end-users, design thinking, emerging tech assessment, etc., that are usually part of an innovation framework.  The key to innovation is not just identifying a pain point, but also to facilitate a different way of thinking in participants to foster active participation from both business and IT stakeholders in coming up with new “out-of-the-box” solutions.
  5. Finally, the conceptual framework illustrated on page 19 can be applied to any industry, not just healthcare.  Putting it differently, the framework does not include anything specific to the healthcare industry but is a fairly general approach.  It is recommended that the framework be adjusted to make it more specific to health.

Reviewer 4 Report

Thank you very much for your paper. In this paper the authors presented a study  to analyze a conceptual framework that can be used in developing innovation projects in hospital. The manuscript deals with an interesting issue and the topic is appropriate for the Journal. The abstract is well-written and clear. Please consider , in the title and abstract the type of the study. In the introduction section the issue is well addressed.

The materials and method section is not clear and appears not well organized. Please reorganized this section because is very confusing.  

Please consider the limits of the study  and check the bibliography according to the journal guidelines

Round 2

Reviewer 1 Report

The manuscript doesn’t add new informations to international literature

Author Response

Dear Reviewers,

Thank you very much for your letter and for the reviewers’ comments concerning our manuscript titled ” Total quality and innovation management in healthcare (TQIM-H) for an effective innovation development: A conceptual framework and exploratory study” (ID: asi-1786127). Those comments are all valuable and very helpful for revising and improving our paper, as well as the important guiding significance to our research. We tried our best to improve the manuscript and made some changes in the manuscript. These changes will not influence the content and framework of the paper.

We appreciate for Reviewers’ warm work earnestly and hope that the correction will meet with approval.

Once again, thank you very much for your comments and suggestions.

Thank you and best regards.

Yours sincerely,

The authors

Reviewer 3 Report

/

Author Response

(The authors gave the same response as above.)

Reviewer 4 Report

The authors have adequately addressed the concerns of this reviewer.

Author Response

(The authors gave the same response as above.)
